# Acute Effects of High-Intensity Interval Training on Diabetes Mellitus: A Systematic Review

**DOI:** 10.3390/ijerph19127049

**Published:** 2022-06-09

**Authors:** Gabriela de Oliveira Teles, Carini Silva da Silva, Vinicius Ramos Rezende, Ana Cristina Silva Rebelo

**Affiliations:** 1Faculty of Physical Education and Dance, Federal University of Goiás, Goiânia 74001-970, Brazil; carinisilva@discente.ufg.br; 2Faculty of Medicine, Federal University of Goiás, Goiânia 74001-970, Brazil; viniciusrezendeef@gmail.com; 3Department of Morphology, Institute of Biological Sciences, Federal University of Goiás, Goiânia 74001-970, Brazil; ana_rebelo@ufg.br

**Keywords:** high-intensity interval training (HIIT), physical exercise, health, hyperglycemia

## Abstract

This study evaluated the scientific evidence on the acute effects of high-intensity interval training (HIIT) on biochemical, cardiovascular, and metabolic parameters in patients with diabetes mellitus. The research took place using two databases (PubMed and Google Scholar) with eligible studies conducted between 2010 and 2020, using the following keywords: (1) high-intensity training/exercise; (2) interval training/exercise; (3) HIIT/exercise; AND “diabetes”. Data extraction was then performed on the eligible studies through content analysis using the categories: author and year of publication; sample characteristics; methods and data collected; intervention protocol; and results found. Methodological quality was assessed using the PEDro scale. Fourteen studies were included, evaluating 168 people with diabetes (122/46 type 2/1) and 42 normoglycemic individuals, which evaluated markers such as capillary and fasting blood glucose, 24-h blood glucose profile, postprandial blood glucose, incidence, and prevalence of hyperglycemia, vascular function and pressure response and control of inflammatory markers. Physical exercise was found to have several acute beneficial effects on the health of the diabetic population, such as reduced capillary and postprandial blood glucose, blood glucose profile, and blood pressure. Moreover, HIIT seems to be a safe and effective alternative in glycemic control and associated factors, superior to continuous moderate-intensity training.

## 1. Introduction

Currently, 463 million people aged between 29 and 70 years live with diabetes, representing 9.3% of the population, and about half of them do not know they have the disease [1]. In many cases, symptoms take a long time to manifest or be noticed by patients, making diabetes mellitus (DM) the silent disease of the 21st century. Brazil has the fifth highest prevalence of the disease, with close to 16.8 million cases. It is among the ten countries with the highest expenditure on health-related costs attributed to diabetes, and the cost to public health increases according to the duration of the disease and the presence of microvascular and macrovascular complications [2].

Chronic microvascular and macrovascular complications related to persistent hyperglycemia are associated with increased morbidity, reduced quality of life, and increased mortality rate. Diabetes treatment is based on pharmacotherapy, adequate nutrition, and physical exercise [3,4]. An intervention with physical training can promote cardiorespiratory and metabolic adjustments and adaptations that can delay the progression of diabetes and improve its prognosis [5,6].

Lack of time is one of the most common barriers to regular physical exercise by people with diabetes [7]. Given the low adherence and concerns about safe and effective training, several types of exercises have been proposed for this population. In recent decades, high-intensity interval training (HIIT) has gained prominence for requiring less time and presenting better responses in endothelial function [8], increasing functional capacity [9], body composition [10], and in improving quality of life [11] than traditional training, such as using continuous moderate intensity. HIIT is a training protocol that alternates short periods of intense exercise (>85% of the VO_2_ max. or maximal heart rate) with short periods of passive or active resting [4]. In people with type 2 diabetes (DM2), publications show that HIIT effectively improves glycemic control, glycated hemoglobin, and cardiorespiratory fitness [12,13].

Some studies have compared the effects of HIIT with continuous moderate-intensity training (CMIT) on indicators of cardiovascular risk in people with diabetes. De Nardi et al. (2018) compared the effects of HIIT with CMIT on functional capacity and cardiometabolic markers in people with DM2 (maximum oxygen volume; glycated hemoglobin; blood pressure; lipid profile; body mass index; and waist-to-hip ratio). Through a systematic review and meta-analysis, the authors concluded that HIIT promotes more significant benefits to functional capacity as evidenced by the 3.02 mL/Kg/min increase (confidence interval 1.42–4.61) in VO_2_ max. in this population. The study did not find differences between the modalities for the glycemia and blood pressure variables or in the other cardiometabolic markers [4].

Considering the growing range of studies on HIIT and the concern over the efficacy and safety of protocols, understanding the effects on the physiological markers of health in this population is necessary. Therefore, the present study analyzed literature concerning the acute effects of HIIT on biochemical, cardiovascular, and metabolic parameters in patients with diabetes.

## 2. Materials and Methods

This systematic literature review study was performed according to the guidelines of the Preferred Reporting Items for Systematic Reviews and Meta-analysis (PRISMA) [14]. The search was performed in two databases (PubMed and Google Scholar) with eligible studies conducted between 2010 and 2020, written in Portuguese or English. The following keywords were used in the surveys: (1) high-intensity training/exercise; (2) interval training/exercise; (3) HIIT/exercise; AND “diabetes”. The protocol for this study was registered in the PROSPERO Systematic Review Protocols Registry under ID: CRD42021250255.

The title and abstract of the articles found were read to verify the presence of keywords. Then, the articles that met the eligibility criteria were separated for a full reading. After selecting the eligible studies, data were extracted in a standardized way, and a database was created in the Excel 2016 (Microsoft, Redmond, WA, USA) platform containing the information from each study. This process was carried out by two researchers independently. Any disagreements were resolved by consensus or by the participation of a third researcher.

### 2.1. Eligibility Criteria

The criteria were established according to the PICOS approach (Population, Intervention, Comparator, Outcomes, and Study Design). In addition, a minimum score of 6 in the assessment of methodological quality using the PEDro scale was accepted as a criterion.

#### 2.1.1. Population

The present review included studies involving adults diagnosed with type 1 or 2 diabetes, aged over 18 years, of both sexes, who were not included in any regular exercise program.

#### 2.1.2. Intervention

Studies reporting the acute effect of HIIT, comparing or not with CMIT on biochemical, cardiovascular, and metabolic parameters in patients with diabetes, were included. Studies had to assess the acute (one session) effect and present the results to be included.

#### 2.1.3. Comparator

CMIT was considered a comparator of the HIIT. The included studies could therefore have a control group or a group that performed CMIT.

#### 2.1.4. Outcomes

The primary outcome of this review is glycemic control, given the importance of this marker for the health of patients with diabetes. Other outcomes were expected, including cardiovascular and metabolic markers such as blood pressure, lipid profile, inflammatory indicators, and C-reactive protein.

### 2.2. Study Design

Randomized clinical trials and case studies were considered, involving a HIIT session.

### 2.3. Data Extraction

The extracted data were entered into an Excel spreadsheet (2016) (Microsoft, Redmond, WA, USA) according to the eligibility criteria using pre-established categories, such as author and year of publication; sample characteristics; methods and data collected; intervention protocol; and results found.

### 2.4. Evaluation of Methodological Quality

The methodological quality of the included studies was assessed using the PEDro scale translated into Portuguese, and the minimum score used was 6 [15].

## 3. Results

The flowchart describing the search and selection of studies is shown in Figure 1. After reading the articles in full, 14 were included in the total that met all the criteria.

Of the 14 studies evaluated, all performed a maximal exercise test before the intervention to diagnose possible exercise-related problems or to indicate session intensity. In addition, all studies performed an aerobic training session, one of which compared aerobic training with resistance training, and 64% of the studies had a control group (no exercise) in the comparison. Eight studies (57%) evaluated the effect of exercise in the postprandial period, seven studies used a standardized diet for participants, and one used online food control. As for the methods, five studies used continuous glucose monitoring (24 h or 32 h) to verify values such as postprandial mean and peak glucose, in addition to the incidence and predominance of hyperglycemia.

The methodological quality of the studies is reported in Table 1, with a minimum score of 6. The characteristics of the study participants are shown in Table 2. Table 3 contains the characteristics of the training methods and protocols of the included studies. Finally, the results are shown in Table 4.

## 4. Discussion

This study reviewed in the literature the acute effects of HIIT on biochemical, cardiovascular, and metabolic parameters in patients with diabetes. It was found that that HIIT improves capillary blood glucose levels and factors associated with cardiovascular risk in people with diabetes, emphasizing the glycemic profile, blood pressure, vascular function, and inflammatory indicators when compared to CMIT or control groups.

We also want to clarify that the results obtained in studies that did not evaluate a control group should be reviewed, and we suggest further studies on the topic.

### 4.1. Capillary Blood Glucose Levels

There is a consensus on the beneficial effect of physical exercise on glycemic control in diabetics. Molecular, physiological and metabolic mechanisms related to glycemic control, induced by exercise, generate several adaptations, including increased concentration and translocation of GLUT-4 in the plasma membrane, improved uptake of muscle glucose, in addition to the increase in muscle fibers that are more sensitive to insulin, and the activity of glycolytic and oxidative enzymes [12].

Santiago et al. (2017) found a reduction in capillary blood glucose of 27.4% after MTCT and 26.9% after HIIT [24]. Mendes et al. (2019) assessed blood glucose every 10 min after the exercise session and found that the hypoglycemic effect lasted for 50 min and that it was greater using HIIT [20]. It is important to note that the authors used a protocol of lower intensity than the others, being 70% of the HRR, while other researchers define HIIT as >85% of the HRM. Viana et al. (2019) demonstrated that HIIT, whether guided by HR or by SPE, promotes a more expressive reduction in blood glucose than CMIT [26].

There is no consensus on the factors leading to a greater reduction in blood glucose by HIIT compared to continuous exercise. However, the high levels of muscle fiber recruitment, the use of muscle glycogen, and the increased insulin sensitivity previously observed during HIIT may increase exercise-induced muscle glucose uptake during and after [29].

### 4.2. Postprandial Blood Glucose and Time/Exposure to Hyperglycemia

Gillen et al. (2012) assessed through continuous glucose monitoring that HIIT was effective in glycemic control three hours after eating, in peak glucose, in postprandial mean, and in hyperglycemia time [27]. Karstoft et al. (2014) compared the effects of HIIT with the CMIT and found a reduction in postprandial blood glucose, more expressive after HIIT [18]. The physiological mechanisms involved in reducing postprandial blood glucose after HIIT are still controversial, but it is possible to speculate that the increased insulin sensitivity in skeletal muscle, the high degree of muscle fiber recruitment, or the use of glycogen may be involved. Furthermore, insulin-independent mechanisms associated with HIIT, such as GLUT-4 translocation and increased endothelial function for up to 72 h, seem to contribute to this effect [27].

Terada et al. (2016) performed a HIIT and CMIT training session with people with diabetes after fasting and after breakfast and verified the glycemic behavior in the following 24 h using continuous glucose monitoring. The authors concluded that when training was performed after fasting, postprandial blood glucose values were lower than with training performed after breakfast [28]. Other authors who investigated the hypoglycemic effects of fasting training concluded that the practice after an overnight fast and before using fast-acting insulin helps to maintain glycemic stability, regardless of the mode or intensity of exercise. This means that the patient does not need to consume carbohydrates to avoid hypoglycemia during training [25]. However, the authors of this study did not evaluate long-term protocols. Further studies are needed to investigate whether regular fasting exercise improves long-term glycemic control.

Metcalf et al. (2018) found that a training session reduced time and the prevalence of hyperglycemia, being more expressive in HIIT than in CMIT [22]. On the other hand, Jayawardene et al. (2017) found different results when verifying that HIIT promoted higher glycemic levels and greater exposure to hyperglycemia than CMIT [17], and this may be attributed to greater cellular stress and subsequent vascular dysfunction. There is also evidence of a strong correlation between postprandial hyperglycemia and the risk of adverse cardiovascular events [22]. Thus, physical exercise seems to have a cardioprotective effect in the diabetic population, regardless of the protocol.

### 4.3. 24 h Glycemic Profile and Hypoglycemia

Using continuous glucose monitoring, Gillen et al. (2012) showed that a single HIIT session reduced mean 24 h glucose and did not increase the risk of hypoglycemia [27]. Metcalfe et al. (2018) compared the acute effects of HIIT and CMIT, and found that CMIT promoted a greater effect on the 24 h glycemic profile than the HIIT protocols [22]. Greater sensitivity to insulin throughout the body can explain this prolonged glycemic control, which is observed immediately after exercise and persists for up to 96 h [30].

There is a need for discussion regarding the risk of hypoglycemia following high-intensity exercise in this population. However, this risk decreases with good professional follow-up, adequate control of variables before, during, and after exercise, and a balanced diet. Scott et al. (2019) and Terada et al. (2016) found that a training session of several protocols did not increase the incidence of fasting hypoglycemia in the following 24 h or at night in diabetics [25,28]. These last authors also found that HIIT promoted a greater reduction in nocturnal and fasting blood glucose than CMIT. Rooyackers et al. (2017) concluded that HIIT decreases the symptoms of hypoglycemia in normoglycemics, and attenuates the cognitive dysfunction induced by hypoglycemia in diabetics [23].

HIT appears to be superior to CMIT in several biochemical aspects in patients with diabetes, but there are still controversies regarding the superiority of one protocol over the other in the 24 h glycemic profile and hypoglycemia; as a result, further studies need to be performed.

### 4.4. Blood Pressure and Vascular Function

In addition to improving glycemic control, reducing cardiovascular risk also has positive implications for DM2 morbidity, mortality, and health care costs. For example, a 2.1/0.9 mmHg decrease in blood pressure (BP) resulted in a 10% reduction in major cardiovascular events in DM2 [31] patients.

Santiago et al. (2017) evaluated BP after one HIIT session and one CMIT session and found a blood pressure reduction using both protocols, being more expressive 30 min after training [24]. Viana et al. (2019) evaluated the BP for 24 h following exercise and found a reduction in this resulting from HIIT guided by the subjective perception of exertion, but not from CMIT, and in the HIIT it was suggested by the participant’s heart rate [26]. The type of exercise also influences this BP control. Francois et al. (2015) found that HIIT cardio training (training based on stimulating the cardiovascular system to gain physical endurance) increased arterial dilation one hour after training, and that HIIT resistance training improved endothelial function in both diabetics and normoglycemics [13].

The mechanisms that guide BP variation during and after exercise are related to hemodynamic, humoral, and neural factors [26,32]. Increased blood flow to skeletal muscles during physical exercise increases the stress on the vascular wall, inducing a greater release of nitric oxide and, consequently, vasodilation. With this decrease in peripheral vascular resistance, there is a drop in BP [32].

### 4.5. Hormones and Indicators of Cellular and Systemic Inflammation

Karstoft et al. (2014) found an increase in lactate and glucagon after HIIT. In 2016, the same authors found a more significant increase in lactate using HIIT over CMIT in a similar study, and that lipid oxidation increased after exercise, regardless of the protocol. Rooyackers et al. (2017) also found a significant increase in lactate and counterregulatory hormones such as adrenaline, norepinephrine, GH, and cortisol during HIIT. Jayaeardene et al. (2017) found that, during HIIT, there was an increase in blood ketone and counterregulatory hormones (epinephrine, norepinephrine, and cortisol) [17].

Several inflammatory markers are associated with exercise. For example, Robinson et al. (2015) evaluated sedentary individuals at high risk of type 2 diabetes, concluding that CMIT led to a reduction in fasting glucose and in the expression of Toll-like receptor 4 (TLR4), which is an inflammatory marker associated with cardiometabolic risk factors, such as insulin resistance and atherosclerosis, while HIIT did not show such effects [33]. These findings provide preliminary evidence that moderate-intensity exercise may lead to greater anti-inflammatory responses in overweight or obese inactive individuals. In addition, Durrer et al. (2017) found similar effects through HIIT, by subjecting diabetic and normoglycemic patients to a HIIT session and observing a reduction in tumor necrosis factor α (TNFα), which is a group of cytokines associated with several metabolic conditions such as hypertension, dyslipidemia, obesity, and insulin resistance, and TLR2, immediately after and one hour after training [16].

Therefore, it has been demonstrated that physical exercise has anti-inflammatory properties, not only in the adipose tissue, but also impacting the phenotype of immune cells and altering systemic inflammatory mediators such as cytokines and interleukins [34].

## 5. Conclusions

Physical exercise has several acute beneficial effects on the health of the diabetic population, including improvement in capillary and fasting blood glucose, blood glucose profile, postprandial blood glucose, decreased incidence and prevalence of hyperglycemia, improved vascular function and response pressure, and control of inflammatory markers.

Thus, HIIT appears to be a safe and effective alternative for glycemic control and associated factors in diabetics. In addition, most studies comparing HIT with CMIT found better acute effects of HIIT on glycemic control and blood pressure. HIIT could have an effect on inflammatory biomarkers. There is likely to be a relationship between changes in inflammatory profile and fat loss. A controlled diet may be a good complement to reduce the inflammatory profile. Further studies are required to determine whether HIIT is a better, worse or an equivalent alternative to medium-intensity aerobic exercise to improve the inflammatory profile. Based on the results obtained in this study, we also sugest the combination of both protocols. For low physical fitness or high-risk patients, starting training programs with continuous moderate-intensity exercise with the gradual inclusion of HIIT sessions using passive recovery is advised. Larger randomized controlled trials of longer duration than those included in this meta-analysis are required to confirm these results.

Furthermore, the studies evaluated here included patients with type 1 and type 2 diabetes, so there is a limitation when analyzing the impact of HIIT on cardiometabolic parameters, regardless of the type of diabetes, given that type 1 and 2 have different genesis and mechanisms.

## Figures and Tables

**Figure 1 ijerph-19-07049-f001:**
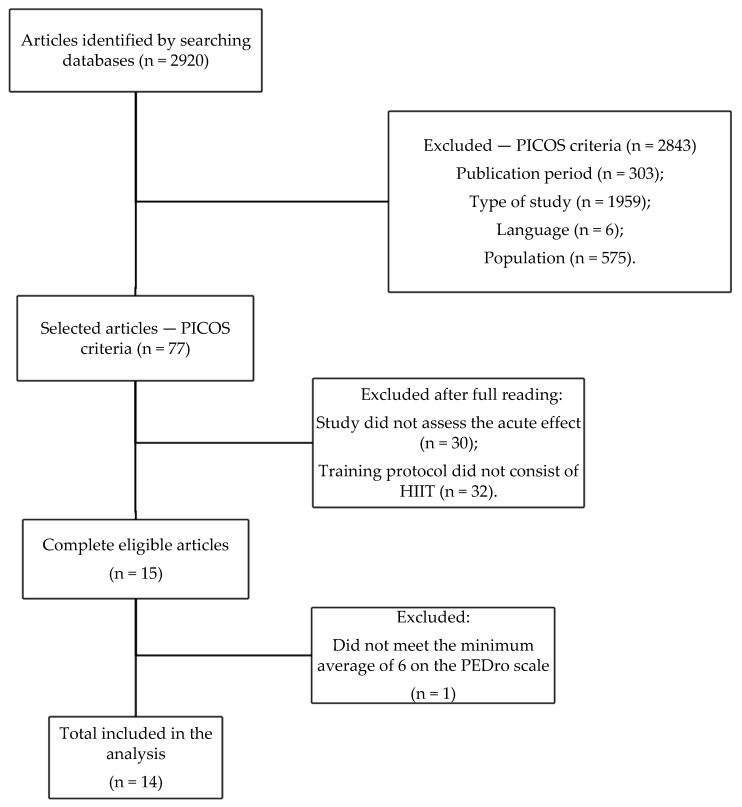
PRISMA flowchart for the search and selection of studies.

**Table 1 ijerph-19-07049-t001:** PEDro score for the included studies.

Reference	Specific Eligibility Criteria	Randomized Allocation	Secret Allocation (Blind)	Groups Similar to Baseline	Blind Participation of Subjects	Blind Intervention and Evaluation	<15% Loss	Intent-to-Treat Analysis	Intergroup Statistics Reported	Measurements and Precision and Variability	Total PEDro Score
Durrer et al., 2017 [16]	1	0	1	0	0	0	1	1	1	1	6
Francois et al., 2015 [13]	1	1	1	1	0	0	1	1	1	1	8
Jayawardene et al., 2017 [17]		1	1	1	0	0	1	1	1	1	8
Karstoft et al., 2014 [18]	1	1	1	1	0	0	1	1	1	1	8
Karstoft et al., 2016 [19]	1	1	1	1	0	0	1	1	1	1	8
Mendes et al., 2019 [20]	1	1	1	1	0	0	1	1	1	1	8
Mendes et al., 2013 [21]	0	1	1	1	0	0	1	1	1	1	7
Metcalfe et al., 2018 [22]	1	1	1	1	0	0	1	1	1	1	8
Rooijackers et al., 2017 [23]	1	1	1	1	0	0	1	1	1	1	8
Santiago et al., 2017 [24]	1	1	1	0	0	0	0	1	1	1	6
Scott et al., 2019 [25]	1	1	1	1	0	0	1	1	1	1	8
Viana et al., 2019 [26]	1	1	1	1	0	0	1	1	1	1	8
Gillen et al., 2012 [27]	1	0	0	1	0	0	1	1	1	1	6
Terada et al., 2016 [28]	1	1	1	1	0	0	1	1	1	1	8

**Table 2 ijerph-19-07049-t002:** Characteristics of the participants in the studies.

Author/Year	Participants	Sex	N	Age (Years)	BMI (kg/m²)
Durrer et al., 2017 [16]	DM2 and NG	5F 5M 5F 4M	10 DM2 and 9 NG	57.9 ± 5.4 and 55.8 ± 9.0	34.8 ± 5.9 and 24.8 ± 3.6
Francois et al., 2015 [13]	DM2 NG TN NG SD	21F 14M	12 DM2 11 NG TN 12 NG SD	57.5 ± 5.0 55.3 ± 9.1 55.1 ± 7.0	35.0 ± 7.0 26.0 ± 5.0 23.0 ± 3.0
Jayawardene et al., 2017 [17]	DM1	9F 3M	12 DM1	40.0 ± 13.0	25.3 ± 3.2
Karstoft et al., 2014 [18]	DM2	3F 7M	10 DM2	60.3 ± 2.3	28.3 ± 1.1
Karstoft et al., 2016 [19]	DM2	3F 7M	10 DM2	60.3 ± 2.3	28.3 ± 1.1
Mendes et al., 2019 [20]	DM2	8F 7M	15 DM2	60.2 ± 3.1	29.6 ± 4.61
Mendes et al., 2013 [21]	DM2	6F 6M	12 DM2	58.7 ± 5.3	30.7 ± 5.6
Metcalfe et al., 2018 [22]	DM2	11M	11 DM2	52.0 ± 6.0	29.7 ± 3.1
Rooijackers et al., 2017 [23]	DM1–AH DM1–AH NG	4F 6M 5F 5M 5W 5M	10 DM1 NAH 10 DM1 IAH 10 NG	23.9 ± 4.4 25.7 ± 5.8 25.2 ± 5.5	23.0 ± 2.3 23.4 ± 1.4 22.5 ± 1.8
Santiago et al., 2017 [24]	DM2	No Information	14 DM2	63.6 ± 9.8	30.3 ± 4.4
Scott et al., 2019 [25]	DM1	8F 6M	14 DM1	26.0 ± 3.0	27.6 ± 13.0
Viana et al., 2019 [26]	DM2	9F 2M	11 DM2	52.3 ± 3.0	28.4 ± 1.5
Gillen et al., 2012 [27]	DM2	No Information	7 DM2	62.0 ± 3.0	30.5 ± 1.9
Terada et al., 2016 [28]	DM2	2F 8M	10 DM2	60.0 ± 6.0	30.8 ± 5.4

DM2, type 2 diabetes mellitus; DM1, type 1 diabetes mellitus; NG, normoglycemic; TN, trained; SD, sedentary; AH, arterial hypertension; NAH, normal awareness of hypoglycemia; IAH, impaired awareness of hypoglycemia; N, number of participants; BMI, body mass index; F, female; M, male.

**Table 3 ijerph-19-07049-t003:** Study training methods and protocols.

Author/Year	Objectives	Methods	Equipment	Type of Training	Duration (min)	Protocol
Durrer et al., 2017 [16]	To determine the impact of a single HIIT session on cellular, molecular, and circulating markers of inflammation in individuals with DM2.	Four familiarization sessions. Evaluation of HR and SPE during, and blood collection before, after, and 1 h after, training 4 h after eating.	Bicycle	HIIT	19	4 min warm-up at 30 W 7 × 1 min at 85% Wmax × 1 min rest at 15%, with 1 min recovery
Francois et al., 2015 [13]	To examine the effect of a single resistance interval aerobic exercise session compared to an equivalent control on endothelial function in untrained and normoglycemic trained participants with DM2.	Six familiarization sessions. Standardized diet 4 h before. Collection: pre, 5 min, 1 h, 2 h post. Collection of flow-mediated dilatation (endothelial function) by ultrasound: 1 min pre, 30 s before dilatation, 3 min during.	Bicycle and three resistance exercises for lower limbs	CYCLE ERGOMETER HIIT RESISTED HIIT CONT	14 14 0	CYCLE ERGOMETER HIIT: 7 × 1 min at 85% VO_2_ max. × 1′ recovery 15%. RESISTED HIIT: 7 × 1 min, in a maximum number of repetitions, of six-legged exercises. CONT: 20 min seated.
Jayawardene et al., 2017 [17]	Examine the effectiveness of a closed-loop system to prevent hypoglycemia and maintain glucose in the target range for adults with type 1 diabetes performing HIIT vs. CMIT, and secondarily investigate exercise-related metabolic changes in blood glucose, ketones, and lactate during the cycle, and to evaluate the association of changes in these parameters with changes observed in the levels of counterregulatory hormones.	1 to 4 weeks between workouts, standardized breakfast.	Bicycle	HIIT CMIT	45	5 min warm-up at 25% VO_2_ max.; HIIT: 6 × 4 min between AT and VO_2_ max., 2 min rest CMIT: 40 min at 70% of AT
Karstoft et al., 2014 [18]	Determine the effect of an interval walking session vs. a continuous walking session equivalent in time and oxygen consumption for glycemic control in subjects with DM2.	1–2 weeks of familiarization. Pause on anti-diabetes medication, exercise, and alcohol. Online food reminder. Blood for glucose and lactate—using HIIT, before, during and after. Using CMIT: (every 15 min). Borg scale and HR. After: 4 h glucose tolerance test, continuous monitoring 32 h after.	Treadmill	HIIT CMIT CONT	60	HIIT: 3 min at 54% and 3 min at 89% VO_2_ max.; CMIT: 73% of VO_2_ max.; CONT: seated.
Karstoft et al., 2016 [19]	Compare the acute effects of interval exercise vs. continuous equivalents in time and oxygen consumption in EPOC, rate substrate oxidation, and lipid metabolism in the hours following exercise in subjects with DM2.	1–2 weeks between tests, 24 h food recall. Direct calorimetry 30′ post-training, and indirect calorimetry for 4 h post. Blood collection before, during, and after, urine 2×	Treadmill	HIIT CMIT CONT	60	HIIT: 3 min at 54% and 3 min at 89% of VO_2_ max.; CMIT: 73% of VO_2_ max.; CONT: seated.
Mendes et al., 2019 [20]	Compare the acute effects of HIIT vs. CMIT on glycemic control in middle-aged and elderly patients with DM2.	Standardized breakfast, 1 week between sessions, postprandial. Capillary blood glucose in the ear, pre, every 10′ and post. Capillary glucose before, during (every 10′), and up to 50 min after (50, 60, 70, 80, and 90).	Treadmill	HIIT CMIT CONT	40	5 min of warm-up HIIT: 5 × 3 min at 70% HRR + 3 min at 30%, 5 min cool down; CMIT: 30 min at 50% HRR, 5 min cool-down; CONT: seated.
Mendes et al., 2013 [21]	To analyze the acute effects of HIIT on postprandial glucose levels in DM2 patients.	After breakfast. Collection: resting blood glucose, 0 min before, 10, 20, 30 min during, immediately after, and in recovery 50, 60, 70, 80, and 90. total 11×.	Treadmill	HIIT	40	5 min of warm-up 5 × 3-min walk 70% HRR, × 3 min at 30%, 5 min cool-down.
Metcalfe et al., 2018 [22]	To examine the effect of a single session of high-intensity, reduced-effort interval training (REHIT) on 24 h blood glucose in men with DM2 compared to a no-exercise control using continuous glucose monitoring.	Standardized diet, continuous 24 h glucose monitoring.	Bicycle	HIIT REHIT CMIT CONT	25 10 30 0	HIIT: 25 min, 10 × 1 min at ~90% HRM; REHIT: 10 min − 2 × 20 s all-out sprints in min 2′40 and 6′40. CMIT: 30 min at 50% HRM; CONT: no exercise;
Rooijackers et al., 2017 [23]	To investigate the effect of HIIT on hyperglycemic symptoms, counterregulatory hormone response, and cognitive function during subsequent hypoglycemia in patients with DM1 and normal awareness of hypoglycemia (NAH) and impaired awareness of hypoglycemia (IAH), but also in healthy participants.	Cognitive function test 15 min after hypoglycemia (attention and memory; verbal fluency; information processing speed). Symptom score and glycemia; 20.40, and 60 min hypoglycemia.	Bicycle	HIIT	15	4 min warm-up at 50 w; 3 × 30 s sprint (as fast as possible); 4 min recovery at 50 W.
Santiago et al., 2017 [24]	To compare acute glycemic and pressure responses of continuous aerobic exercise with interval aerobic exercise in patients with DM2.	Blood glucose and BP collection: pre, immediately post, 5, 10, 15, 20, 25, 30 min. Food control.	Treadmill	HIIT CMIT	45 35	HIIT: 45 min, 9 × 5 min (4 min 85–90% AT + 1 min < 85% AI); CMIT: 35 min between 85 and 90% HR of AI.
Scott et al., 2019 [25]	Compare the effects of a single session of HIIT with a session of CMIT on glucose concentrations in the subsequent period of 24 h.	Standardized diet with 3 meals, 3-day food recall, continuous monitoring of glucose in the abdomen 24 h. Fasting training. Pre and post glucose.	Bicycle	HIIT CMIT CONT	17 30 0	5 min warm-up at 50 W; HIT: 17 min, 6 × 1 min at 100% VO_2_ max. × 1 min rest; CMIT: 30 min at 65%. VO_2_ max.; CONT: no exercise.
Viana et al., 2019 [26]	Test the hypothesis that (1) SPE is as efficient a tool as HR relative to cardiopulmonary testing to guide and self-regulate HIIT; (2) metabolic and hemodynamic responses of HIIT are superior to CMIT, regardless of whether suggested and regulated by SPE or HR relative to the cardiopulmonary test.	Blood glucose, HR, BP, femoral pulse velocity. and endothelial reactivity before, after, and 45 min after.	Treadmill	HIIT SPE HIIT HR CMIT CONT	25 25 30 0	4 min warm-up to 9 SPE or 50% HRR; HIIT SPE: 25 min, 21 min being 1 min at 15–17 SPE + 2 min at 9–11 SPE; HIIT HR: 25 min, 21 min being 1 min at 85% HRR + 2 min at 50%; CMIT: 30 min, 26 min at 11–14 SPE; CONT: 30 min seated.
Gillen et al., 2012 [27]	Examine the glycemic response 24 h after a HIIT session consisting of cycling efforts of 10 × 60 s at ~90% of HRM, interspersed with 60 s of rest.	Continuous monitoring of glucose 24 h, standardized diet, training 1.5 h after breakfast. Collection: 24 h glucose mean, hyperglycemia time, 3 h post-eating, glucose peak, and postprandial mean 60 min to 120 min.	Bicycle	HIIT	25	3 min warm-up at 50 W; 10 × 1 min at 90% of HRM and 60 s rest; 2 min back to calm at 50 W.
Terada et al., 2016 [28]	To compare the acute glycemic response of a HIIT and CMIT session performed under fasting and postprandial conditions.	24 h continuous monitoring of blood glucose, 48 h between workouts. 24 h average, postprandial, fasting, nocturnal, variability, and time in hypoglycemia and hyperglycemia.	Treadmill	HIIT fasting HIIT with coffee CMIT fasting CMIT with coffee CONT	60	HIIT: 3 min at 40% + 1 min at 100% of VO_2_ max. (15×); CMIT: 55% of VO_2_ max. CONT: no exercise.

HIIT, high-intensity interval training; CMIT, continuous moderate-intensity training; CONT, control; MIN, minute; blood pressure; HR, heart rate; HRM, heart rate maximum; HRR, heart rate reserve; SPE, subjective perception of effort; AT, anaerobic threshold; W, watts; Wmax, maximum watts; VO_2_ max., maximum oxygen volume; REHIT, reduced exertion high-intensity interval training; EPOC, excess post-exercise oxygen consumption.

**Table 4 ijerph-19-07049-t004:** Results of the studies.

Author/Year	Results
Durrer et al., 2017 [16]	HIIT reduces TLR2 expression after and 1 h after exercise in DM2 and normoglycemics.
Francois et al., 2015 [13]	HIIT resistance exercise is efficient to improve the endothelial function of DM2 in trained and sedentary normoglycemics.
Jayawardene et al., 2017 [17]	HIIT resulted in higher glycemic levels and greater hyperglycemic exposure than CMIT during training. There was a greater increase in ketone levels in HIIT than in CMIT. Elevation of counterregulatory hormones (epinephrine, norepinephrine, and cortisol). GH and glucagon did not change.
Karstoft et al., 2014 [18]	HIIT improves postprandial glycemic control in DM2 compared to CMIT.
Karstoft et al., 2016 [19]	EPOC was higher after HIIT compared to CMIT. Lipid, carbohydrate, and protein oxidation did not differ. HR, SPE, and VO_2_ were similar. Lactate was higher at HIIT. Lipid oxidation increases during and after exercise in DM2, but with no difference between protocols.
Mendes et al., 2019 [20]	Both workouts reduced blood glucose during exercise and within 50 min of recovery. The effect of HIIT was greater than that of CMIT.
Mendes et al., 2013 [21]	Capillary blood glucose significantly different at 20, 30, 40 min during, and 50 min after exercise. HIIT appears to be an effective and safe strategy for acute glucose control in DM2 patients.
Metcalfe et al., 2018 [22]	Hyperglycemia time was reduced in all protocols, being more expressive in HIIT. CMIT promoted the greatest beneficial effect on the 24 h profile. REHIT reduced the mean 24 h glucose and the prevalence of hyperglycemia compared to the control. REHIT may offer an efficient option to improve the glycemic profile in males with DM2.
Rooijackers et al., 2017 [23]	HIIT reduced symptoms of hypoglycemia in normotensive individuals, but not in healthy or hypertensive individuals. HIIT attenuated the cognitive dysfunction induced by hypoglycemia.
Santiago et al., 2017 [24]	Blood glucose reduced immediately after and during recovery in both protocols, being more expressive in CMIT. SBP reduced in both, with greater reduction within 30 min of recovery. Both were effective in reducing blood glucose and acute blood pressure in patients with DM2.
Scott et al., 2019 [25]	There was no difference between HIIT and CMIT in the 24 h glycemic profile. Fasting training did not increase the incidence of 24 h or nocturnal hypoglycemia. Stable glycemic control during training.
Viana et al., 2019 [26]	HIIT was more effective than CMIT in lowering blood glucose regardless of which was used. Only HIIT SPE reduced BP 24 h.
Gillen et al., 2012 [27]	There was a reduction in mean glucose 24 h, hyperglycemia time, 3 h after eating, peak glucose, and mean postprandial 60 min to 120 min after HIIT. HIIT promotes improved glycemic control in people with DM2.
Terada et al., 2016 [28]	Fasting exercise reduced postprandial blood glucose more than after breakfast. HIIT promoted a greater reduction in nocturnal and fasting blood glucose than CMIT. Compared to control, fasting, HIIT improved glycemic parameters. There was no increased risk of hypoglycemia.

DM2, diabetes mellitus type 2; HIIT, high-intensity interval training; CMIT, moderate-intensity continuous training; TLR2, Toll-like receiver 2; GH, growth hormone; BP, blood pressure; SBP, systolic blood pressure; HR, heart rate; SPE, subjective perception of effort; EPOC, excess post-exercise oxygen consumption; REHIT, reduced exertion.

## Data Availability

Data available on Google Drive. The data presented in this study are openly available at: https://drive.google.com/drive/folders/1apeVEx9Wc-wnV_oxabUzcKOTyE-po4jY?usp=sharing. Accessed on 13 February 2022.

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
