# Peer review of "Acute Effects of High-Intensity Interval Training on Diabetes Mellitus: A Systematic Review"

_ijerph, 2022, doi:10.3390/ijerph19127049_

Round 1
Reviewer 1 Report
The article addresses an interesting topic. However, it is necessary not to demonstrate a wrong concept and establish what really is high-intensity interval training?
In the third paragraph of the introduction, you refer to HIIT as "high-intensity interval aerobic training". The HIIT is a training protocol that alternates short periods of intense exercise (>85% of the VO2max. or maximal heart rate) with short periods of passive or active resting. Due to these intensities, this protocol is not a predominantly aerobic exercise; it is a predominantly anaerobic exercise. So, High-intensity interval aerobic training is not the correct terminology and may be changed to "High-intensity interval training".
Please establish which protocols are defined for a physical exercise to be considered HIIT in the introduction section.
In the section "Materials and Methods."
why do you use only two databases (PubMed and Google Scholar)? Why did you not use Scopus or other databases also?
Did you consider asking experts in the field if they knew any article on the subject that was not included in your selection?
In the "Eligibility Criteria" section, the CMIT is written again in full; please use only the abbreviation.
Table 3 - in the study of Francois et al. 2015, what is Aerobic HIIT? You have defined the exercise by the wrong physiological characteristics. I suggest changing HIIT performed in ergometers (in this case, a cycle ergometer). Also, in this article, bouts of 1 min, at an all-out number of repetitions, of six legs exercises were used. This training structure should be mentioned in the table.
What does anaerobic injunction mean? Please clarify!
The articles of Mendes et al., 2019 and Mendes et al., 2013, not use a true HIIT protocol. The intensity used in these articles (70% of the reserve heart rate)is significantly lower than the protocols established for HIIT (>85% VO2max. or maximal heart rate). This intensity may be considered in the discussion session or should be removed Mendes et al. articles from this review.
In table 3, in Rooijackers et al. 2017 article section, please place the watts used in the bicycle for 30s sprints.
Discussion
The results obtained in articles that did not use a control group should be a caveat in the discussion section.
Author Response
We appreciate all contributions and comments made by the reviewers and editors. We made our best to observe all the considerations and changed the manuscript accordingly. All changes are highlighted in the article and in the file we provide detailed answers.

Reviewer 2 Report
The Authors decided to present potential advantages of HIIT in diabetes patients. This idea is very interesting and worth elaborating, however I would like to express my concern about the study design.
The manuscript does not have lines numbers in the first part, so it’s hard to point specific places to correct.
Authors focused on diabetes type 2 in the introduction. I suggest to make it clear in the article title and in some places in the manuscript (eg. last lines of introduction).
At the same time, the Authors included type 1 diabetes patients into the analysis, so regarding the introduction mentioning only T2D it is unclear with type/types the review refers to. As type 1 diabetes has different mechanism and genesis (e.g. lack of insulin resistance) it is not appropriate to analyze HIIT impact on cardiometabolic parameters regardless of the diabetes type. I suggest to repeat the analysis but without T1D studies as they are not relevant to T2D subjects.
The conclusions also do not match exactly the results presented by the authors. The authors claim that HIIT is a safe and antiinflammatory alternative of CMIT, while at the same time they point the HIIT leads to higher lactate and stress hormones levels. That is why I find the conclusions way too bold in the context of the results. I suggest forming more careful conclusions.
I highly encourage the Authors to revise the design of the study and to submit the manuscript again after the revisions as the analyzed subject is very interesting.
Author Response

(The authors gave the same response as above.)

Reviewer 3 Report
From a methodological point of view, the work is carried out. The results of the work are well reported and discussed. A major problem remains in the conclusion and a fortiori the recommendations. There seems to be a bias for HIIT since not all studies show unique benefits. There seems to be a bias towards HIIT since not all studies show homogenous benefits (e.g.s (e.g.,. : markers of inflammation and many alse in your manurcipt). l 134-134 the effects of CMIT on anti-inflammatory response is cited., line 74-78 on 24h glycemia, and many others various results regardin HIIT and CMIT in the discussion are reported. In general, the effects of HIIT are not unanimous and unilateral, and the paper shows this well. So, the writing of the conclusion and the abstract should be reviewed to be consistent with the review which is of good quality. You should avoid "recommendations" that could be understood as prescriptions and cannot be formulated in this type of paper.
Author Response

(The authors gave the same response as above.)

Round 2
Reviewer 1 Report
Congratulations to the authors. Following the reviewers' suggestions, the authors made improvements that put the article in a position to be accepted for publication.
Author Response
Thanks for all the considerations. We appreciate!
Reviewer 2 Report
The Authors decided to present potential advantages of HIIT in diabetes patients. This idea is very interesting and worth elaborating, however I still would like to express my concern about the study design.
The Authors did not change the analyzed group i.e. they still analyzed T1D and T2D altogether. Even though the Authors claim that each type was analyzed with its own particularities, there is not any comment in discussion or conclusions regarding the impact of the type of diabetes (1 or 2) on selected studies results. The only information about that is provided in the table. That is why I still suggest correcting that.
I highly encourage the Authors to revise the design of the study and to submit the manuscript again after the revisions as the analyzed subject is very interesting.
Author Response
we appreciate the consideration and have added a paragraph on the subject at the end of the conclusion.
Reviewer 3 Report
discussion : line7 : "(...) when compared to CMIT/Control " do you mean in the same study or not ?
When you evoque glycemia for example you never mention any value - high or low glycemia cannt be per se considered as a problem, it depends on the values. Could you be more precise to facilitate interpretation ?
Intermediate conclusions are missing for each endpoint as it is quite difficult in some cases to understand the superioriy of HIT. ie 24-hour glycemic profile and hypoglycemia : the studies reported regarding their results do not allow to consider HIT to be of more interest than HIT. What is you position ? You have to express it before the conclusion.
You replace recommend with suggest but the weight of the evidence remains unevaluated and not discussed. Without repeating the studies one by one, you report the effects of HIT but too often you conclude "vs" CMIT even though it is not a comparator in the same studie and quite rarely in the other studies cited. It seems to be more cautious to highlight the effects of HIT without concluding with such certainty on the general superiority of HIT over CMIT.
minor : - p3 Suppress « other » line 40, as lactate in not an hormone. - P2 of discussion line 41 rewrite the sentence : "There is a discussion to be had about the risks of hypoglycemia following high-intensity exercise in this population. "(English) ie There is a need of discussion regarding the risk of hypoglycemia following high-intensity exercise in this population .
Author Response
-
We mean in the general conclusion, studies that compared with CMIT or control. We changed this detail in line 7.
-
When we talk about glycemia, the term improvement or worsening is always followed. Example: "HIT promoted improvements in capillary glycemia levels [...] in the glycemic profile". Therefore, it goes according to each parameter, if capillary blood glucose has improved, it means that it has approached the desired levels.
- Correction was made in line 13 of the thread.
- Correction was made in line 36 of conclusion, and line 6 of the next page. We agree with the reviewer and avoid using "vs" as it is not a methodologically controlled comparison.
- Correction was made. Thanks.